# The Impact of an After-School Physical Activity Program on Children’s Physical Activity and Well-Being during the COVID-19 Pandemic: A Mixed-Methods Evaluation Study

**DOI:** 10.3390/ijerph19095640

**Published:** 2022-05-05

**Authors:** Hilary A. T. Caldwell, Matthew B. Miller, Constance Tweedie, Jeffery B. L. Zahavich, Ella Cockett, Laurene Rehman

**Affiliations:** 1School of Health and Human Performance, Dalhousie University, P.O. Box 15000, Halifax, NS B3H 4R2, Canada; hilary.caldwell@dal.ca (H.A.T.C.); matthew.miller@acadiau.ca (M.B.M.); ctweedie@dal.ca (C.T.); zahavich@dal.ca (J.B.L.Z.); ecockett@dal.ca (E.C.); 2Healthy Populations Institute, Dalhousie University, 1318 Robie Street, Halifax, NS B3H 3E2, Canada; 3School of Kinesiology, Acadia University, P.O. Box 143, Wolfville, NS B4P 2R6, Canada

**Keywords:** pandemic, school, child, youth, thematic analysis

## Abstract

Introduction: This study evaluated the impact of the Build Our Kids’ Success (BOKS) after-school program on children’s physical activity (PA) and well-being during the COVID-19 pandemic. Methods: Program leaders, children, and their parents were recruited from after-school programs in Nova Scotia, Canada, that delivered BOKS programming in Fall 2020. After participating, Grade 4–6 children (*n* = 14) completed the Physical Literacy Assessment for Youth Self (PLAYself), Physical Activity Questionnaire for Older Children (PAQ-C), the Physical Activity Enjoyment Scale (PACES), and 5 National Institutes of Health (NIH) Patient-Reported Outcomes Measures Information System (PROMIS) scales. Children (*n* = 7), parents (*n* = 5), and program leaders (*n* = 3) completed interviews, which were analyzed for themes inductively. Results: The average PAQ-C score was 2.70 ± 0.48, PLAYself was 68.23 ± 13.12, and PACES was 4.22 ± 0.59 (mean ± SD). NIH PROMIS scores were below standard means (cognitive function, family relationships) or within normal limits (peer relationships, positive affect, and life satisfaction). A thematic analysis of interviews revealed that children’s PA levels were impacted by the pandemic and that BOKS positively impacted children’s physical well-being and integrated well with school-based activities. Conclusions: Participation in BOKS provided an overall positive experience and may have mitigated COVID-19-related declines in PA in well-being. The results of this evaluation can inform future physically-active after-school programming.

## 1. Introduction

During the COVID-19 pandemic, parents of Canadian children and youths reported that their children were engaging in less physical activity than before the pandemic [1,2]. Some parents have reported that they felt children were not getting enough heart-pumping, moderate-to-vigorous physical activity (MVPA) during the pandemic [3] Given the established physical, psychological/social, and cognitive health benefits of engaging in physical activity for children and youths [4], the observed changes have potential to impact the current and future health and well-being of children and youths. Reports suggest that the well-being of children and youths is also suffering. In summer 2020, youths in Nova Scotia reported that they missed social and physical activity opportunities (i.e., physical education and extra-curricular sports) while schools were closed and were looking forward to seeing friends and participating in physical activity once schools opened again [5]. Children also reported missing their friends and social interactions while sports and activities were cancelled during the pandemic [3].

The health behaviors and well-being of children and youths during the COVID-19 pandemic varied based on the severity of public health restrictions in different regions [6]. The region of interest for this study is Atlantic Canada, and Nova Scotia, specifically. In summer 2020, almost half of the youth surveyed in Nova Scotia reported they were doing less physical activity than before the pandemic [5], and strategies were needed to address this trend. During the implementation of this study’s physical activity program, Nova Scotia entered its second wave of COVID-19 and further restrictions were implemented province-wide [7]. At this time in Fall 2020, sports and recreation facilities were closed, limiting children’s participation in extra-curricular recreation and sports [7]. Schools remained open and the only structured physical activity available for children and youth was that delivered in school or childcare settings, including before- and after-school programs. The timing of our study provided a natural experiment to determine the impact of an after-school physical activity program delivered during a period of strong public health restrictions. 

Schools provide opportunities for physical activity before, during, and after-school [8]. Before- and after-school programs are often delivered as childcare, but they provide an opportunity for children to be active with their peers in a supervised setting. After-school physical activity programs may be most effective when implemented in schools versus community settings as this is associated with higher attendance and participation [9]. For example, the Build Our Kids Success (BOKS) program has been delivered in before school programs, and participation was associated with increased physical activity levels, physical health, students’ positive affect, and student engagement [10,11]. BOKS was developed by an experienced educational leadership team with a mission to ‘make physical activity and play part of every child’s day’, and it consists of a variety of free, fun, and engaging resources to support children to develop a lifelong commitment to health and wellness. BOKS programming includes full lesson plans, short movement bursts and movement-based games, activities, and resources [12]. Given the positive findings associated with participation in BOKS programming in before-school programs, further investigation of the benefits of BOKS is warranted, particularly in after-school programs, as many Canadian elementary schools utilize after-school programs as a form of childcare, and after-school programs are a promising setting for physical activity promotion [13]. 

We recently reported on the impact of BOKS programming on parent reports of children’s physical cognitive, social, and emotional health during the COVID-19 pandemic. We found that the programming may have had a protective effect on children’s physical activity levels and well-being; however, this was only based on parental perspectives and not children or BOKS trainers [14]. It is important to capture the perspectives of parents and children as both perspectives are valuable, and the parent–child agreement on mental health and quality of life measures is generally low [15,16]. Given the potential impact of after-school physical activity programming on children and youths’ physical, mental, and psychosocial health, the purpose of this study was to determine the impact of BOKS programming in an after-school setting on children’s physical activity and well-being during the COVID-19 pandemic from the perspectives of children, parents, and trainers. 

## 2. Materials and Methods

### 2.1. Study Design

The Dalhousie University Research Ethics Board (REB# 2019-5024) approved this study. The evaluation centered on the Halifax Regional Centre for Education Excel after-school program sites that implemented Build Our Kids’ Success (BOKS) programming in Fall 2020, as reported previously [14,17]. All Excel leaders had been trained on how to implement the BOKS programming, so the quality of the programming was consistent across schools; however, schools were not constrained to specific lesson plans during this evaluation. Although children were registered for Excel for the duration of the semester, not all children were in attendance each day. Inclusion criteria for the study included Excel program leaders, parents, and children in grades 4 to 6 who were participating in at least one session of BOKS per week. Program leaders reported that BOKS programming was delivered, on average, for 70 min per week, with an average of 20 children in attendance per site. Recruitment for all surveys and interviews was done via email invitation from the Excel Management to all families registered in the Excel after-school program at 31 sites. Child participants (*n* = 14) completed digital surveys, and program leaders (*n* = 3), children (*n* = 7), and parents (*n* = 5) completed interviews virtually via Zoom. The number of interview participants was limited by the timing of the study, as the semester had ended and as the BOKS programming was completed, and by the COVID-19 pandemic, as in-person recruitment and data collection were not possible. Program leaders and parent/guardians of child participants provided digitally signed consent via Opinio online software program. Child participants completed an oral assent session at the beginning of online data collection for interviews. Participants were informed that their participation was voluntary and that they could decline to answer any questions throughout the data collection process or withdraw from the study at any time without repercussion.

### 2.2. Quantitative Measures

Participating children and youths completed four quantitative measure questionnaires from home using Dalhousie University’s Opinio online software program in December 2020. As a result of the pandemic, the original evaluation protocol, which included in-person pre- and post-measures of children’s physical literacy and physical activity, was modified to meet the current public health restrictions at the time of assessments. In-person assessments were not feasible, and all included measures were collected virtually.

### 2.3. Physical Activity

Physical activity participation was assessed with the Physical Activity Questionnaire for Children (PAQ-C), a self-report, 7-day recall instrument that measures generate moderate-to-vigorous physical activity during the school year [18]. The PAQ-C is a low cost, valid, and reliable measure to assess physical activity in 8-to-14-year-old children and youths [19,20]. The PAQ-C includes 9 items that are scored from 1 to 5, and the PAQ-C total score is calculated as the average of 9 items, and a score of 1 indicates low physical activity, and a score of 5 indicates high physical activity [18]. 

### 2.4. Physical Activity Enjoyment

The Physical Activity Enjoyment Scale (PACES) was initially developed to assess physical activity-related feelings of enjoyment, boredom, pleasure, challenge, accomplishment, frustration, gratification, and exhilaration in college-aged students, but it has since been validated for use with children and youths [21,22]. The modified PACES for children includes 16 statements that begin with “when I am physically active…” and asks about physical activity-related thoughts and feelings. Items are scored a 5-point Likert scale from “disagree a lot” to “agree a lot”. The seven negatively worded items were reverse scored, and an average score of all 16 items was calculated [21]. A higher score represents a higher physical activity enjoyment score. 

### 2.5. Physical Literacy

Physical literacy was assessed with the Physical Literacy Assessment for Youth Self (PLAYself), a 22-item self-evaluation of a child’s perception of their own physical literacy. Physical literacy is defined as “the motivation, confidence, physical competence, knowledge, and understanding to value and take responsibility for engagement in physical activities for life” [23]. PLAYself includes 4 subsections: environment, physical literacy self-description, relative rankings of literacies (literacy, numeracy, and physical literacy), and fitness. Each item is scored from 0 to 100, and the average of all items generates the total score [24]. PLAYself can be completed independently by children (aged 7 years and older) and demonstrates acceptable internal consistency, test–retest reliability, and validity [25,26]. 

### 2.6. Physical, Mental, and Social Health

Participants completed the National Institute of Health Patient-Reported Outcomes Measurement Information System (NIH-PROMIS) to assess child mental and social health. Five measures of the NIH-PROMIS Pediatric self-report scales were included in this study. Children completed 4-question scales for life satisfaction, family relationships, and positive affect; a 7-question scale for cognitive function; and an 8-item scale for peer relationships, and each question was a 5-point scale [27,28,29,30,31]. PROMIS measures are advantageous for use in pediatric research because they have greater precision, less error, a large range of measurement values, and fewer items than many conventional measures. PROMIS measures have a common metric, the T-Score, with a mean of 50 and a standard deviation of 10, where 50 equals the means for the U.S. general population [32]. 

### 2.7. Quantitative Statistical Analyses

All analyses were conducted in STATA Version 14.2 for Mac. Descriptive statistics (mean, standard deviation, and range) were calculated for each outcome of interest. Scores for PROMIS scales were compared to normative data [32]. 

### 2.8. Qualitative Measures

Semi-structured interviews were conducted with parents/guardians, children (students), and BOKS program leaders. Interviews were held via ZOOM between December 2020 and February 2021 and lasted between 20–30 min. Interviews were performed to understand the impact of the BOKS program from the various participant’s perspectives. Interviews were audio-recorded and later transcribed for analysis. 

Parent/guardians and program leaders were asked questions that aimed to capture their perceptions of children’s experiences of the BOKS program, including enjoyment; participation level; and changes in confidence, behaviour, mood, and attitude toward physical activity as a result of their involvement. Questions also focused on challenges and/or barriers to physical activity participation due to the pandemic. Children were asked similar questions about their experiences. The interview questions were developed by the study’s investigators based on the program evaluation’s research questions. 

### 2.9. Qualitative Analyses

A qualitative analysis was conducted using QSR International NVivo Version R1. Prior to the analysis, recorded interviews were transcribed and identifying information removed. An inductive thematic analysis was conducted using the six-phased approach outlined by Braun and Clarke (2006), which involved (i) reading and rereading (familiarization) the transcripts and note-taking; (ii) generating initial codes; (iii) searching for themes; (iv) reviewing the themes and examining their coherence with the entire dataset; (v) defining and naming the themes; and (vi) producing the report. An inductive approach to coding the data was considered most appropriate, as it occurs without trying to fit the data into a pre-existing coding frame, such as the interview guide, or being driven by the researchers’ theoretical interests. Therefore, generated themes tend to be strongly linked to the data themselves [33]. Analyses were completed by two of the study’s co-authors (J.B.L.Z and E.C.). To ensure rigor in their analyses, the co-authors independently reviewed transcripts and met regularly to review content and determine themes. 

## 3. Results

### 3.1. Quantitative Measures

Fourteen school-age children in Grades 4–6 (average age: 9.25 years; 55% female) completed student questionnaires. The average PLAYself score was 68.23 (out of a maximum 100), PAQ-C was 2.7 (out of a maximum 5), and PACES was 4.22 (out of a maximum 5). NIH-PROMIS raw and adjusted T-scores are included in Table 1. The life satisfaction scores were similar to the standard median, the mean family relationship score was slightly below the fair-good cut-point, cognitive function was below the standard mean, and positive affect and peer relationship scores were similar to the standard medians [32].

### 3.2. Qualitative Measures

Individual interviews with parents/guardians, children, and BOKS program leaders revealed five major themes.

#### 3.2.1. Theme One: Children’s PA Levels Were Impacted by the Pandemic

Throughout the 2020/2021 school year, the BOKS program had been a regular component of the Excel before- and after-school program in Halifax Regional Centre for Education schools in Nova Scotia. Several children indicated that their participation in community-based sports and/or leisure activities were either cancelled or unavailable due to restrictions put in place because of the pandemic. Some program leaders believed that the BOKS program contributed to a large percentage of many children’s total daily time spent being physically active, and for some children, BOKS may have been their only source of regular physical activity during the pandemic. Many of the parent/guardian respondents agreed that their child’s physical activity decreased since the beginning of the pandemic in comparison to pre-pandemic times. One parent recalled that “there was a lack of physical activity just for the fact they couldn’t go anywhere. Parks were limited, soccer was cancelled, even Girl Guides stopped. Everything, everywhere was cancelled!”

One program leader observed disconcerting physical fitness levels amongst students at the beginning of the BOKS program [September 2020]. They described how students were “barely getting through a 30-min session” and tied it to students “being off since March [2020]”. For context, all public schools in Nova Scotia were closed from March–June 2020 due to concerns over the spread of the COVID-19 virus, and they were reopened the following school year in September 2020. The same program leader believed these marked declines in fitness were the result of their parents/guardians having to work from home and being unable to “get organized to get the kids out”.

Although none of the student participants commented directly about their diminished PA levels as a result of the pandemic, some described engaging in creative forms of PA due to public health enforced closures and restrictions. For example, one male student explained how he learned new tricks on his trampoline while competing with his friend next-door who also had a trampoline, with conversations taking place through a backyard fence due to enforced social restrictions.

#### 3.2.2. Theme Two: BOKS Positively Impacts Children’s Physical Well-Being

Both parents/guardians and program leaders attributed children’s participation in BOKS with a positive impact on children’s overall wellbeing, with an emphasis on physical health. Some program leaders reported observed improvements in children’s capacity to perform physical work (e.g., exercising while wearing full outdoor gear), marked increases in muscular strength and endurance (e.g., number of push-ups performed), and enhanced willingness to participate in physical activity.

In addition to the physical benefits of the BOKS program, it was interesting to learn about the children’s preferences and perspectives of the program. For example, one child expressed how much they enjoyed the freedom of choice in workout activities and referred to BOKS as the “body in motion” program. Similarly, another student stated that exercise is “really important and we need it to stay fit”. 

#### 3.2.3. Theme Three: BOKS Participation Positively Impacts Children’s Cognitive and Emotional Health

Most respondents either experienced first-hand (children) or observed (parents/guardians and BOKS program leaders) improvements in children’s mood and behaviour during the pandemic because of BOKS participation during the pandemic. Children reported increased energy levels and felt less tired, while some reported an enhanced ability to concentrate and focus after participating in a BOKS session. Parents/guardians observed improvements in their children’s mood. One parent claimed that their child was less likely to overreact, was noticeably calmer, and was better at handling conflict amongst peers. Another parent commented that her daughter is “definitely happier and more cooperative when she’s been active. She definitely needs that in her life. I need that in my life.”

Not surprisingly, many of the students referenced interruptions to their typical sport and PA participation outside of school as a result of the pandemic and consequently spoke about missing their friends. When the students were asked what they liked most about the BOKS program, all students indicated how BOKS provided them an opportunity to play games, run around, or simply “just play outside” with friends.

One program leader associated participation in the BOKS program with noticeable positive changes in student behaviour. They detailed how their school adopted a modified version of the BOKS performance-based reward system (i.e., Kid of the Week), to account for students’ best effort, positive attitude, and participation. This led to increased engagement and less conflict between children, as more became interested in “earning Kid of the Week status”, which was not limited to just one child being selected.

#### 3.2.4. Theme Four: BOKS Provided an Added Confidence Boost to Children

Most parent/guardian participants admitted that they did not know many details about the BOKS program prior to participating in the study. One parent was pleasantly surprised to learn about the non-competitive nature of BOKS, as it had supported their overtly shy child to participate in the program. One program leader observed marked improvements in children’s confidence, and, in particular, girls: “their confidence certainly went up; they’re far more willing to try something with BOKS than they would in the gym with the other kids, especially for the girls or any boys that are just not physically built for sports or just don’t have that interest”. Interestingly, one program leader observed that children were less motivated to participate in BOKS when administered in smaller groups of less than five children. Another program leader attributed the positive change in children’s confidence to the non-sport-specific nature of BOKS. Similarly, one parent who described their child as “never the athletic type” associated their child’s confidence gains with how inclusive BOKS is for children that perhaps do not have as strong of a connection to sport. After defining confidence to the students using lay terms, most agreed that their participation in BOKS had led them to try new forms of physical activity on their own and allowed them to make new friends as a result.

#### 3.2.5. Theme Five: BOKS Integrated Well with School-Based Activities

Several program leaders indicated how well the BOKS program integrated with their school’s existing programs. One program leader referred to the BOKS program as the “Rolls Royce” of tools, with respect to getting the children moving. Another commented on how easy the lesson plans were to use, particularly the BOKS Bursts sessions. One leader stated that “the BOKS Bursts are great for when we need to get the kids to do something for five or 10 minutes before we transition to our next programming component. The kids love it! We love it!” Another leader described how BOKS was having a positive impact on children’s nutritional habits: “we’re starting to see students who can read, are reading labels now on their snacks, and some of them are even starting to make different choices about the kinds of snacks they’re bringing to school.” Neither the parents nor student participants were asked questions about the integration of the BOKS program with respect to school-based activities.

## 4. Discussion

The purpose of this study was to report on the impact of BOKS programming implemented in an after-school program during the second wave of the COVID-19 pandemic on children’s physical activity and well-being. After participating in BOKS programming, a small sample of participants reported moderate-to-high perceived physical literacy, moderate physical activity, and moderate physical activity enjoyment. The thematic analysis of interviews with children, parents/guardians, and program leaders revealed that the BOKS was one of the only sources of physical activity for children during the COVID-19 pandemic and that children’s physical and cognitive well-being benefited from participating in BOKS programming. In addition, it was observed that BOKS programming integrated well with school-based activities and that the lesson plans were easy to follow and implement for program leaders. Overall, BOKS programming provided a positive experience for children and youth and may have mitigated COVID-19-related declines in physical activity and well-being. 

Several studies have reported on the decline in children’s physical activity, sports participation, and play during the COVID-19 pandemic. During the second wave of the pandemic in Fall 2020, only 14% of children and youth from across Canada met the recommended physical activity guideline [2]. In summer 2020, less than a third of children and youths in Nova Scotia met the recommended physical activity guideline [5]. During the second wave of the COVID-19 pandemic in Nova Scotia (Fall 2020), schools and childcare centers remained open, while access to sport and recreation was limited [34]. Our qualitative interviews revealed that BOKS programming was the only source of structured physical activity for some children and youth at this time. These findings complement the moderate physical activity levels reported by children on the PAQ-C, suggesting that the BOKS programming may have had a protective effect on participants’ physical activity levels. At times when public health restrictions limit physical activity opportunities (i.e., sports), it is essential that physical activity continue to be promoted in those settings where it is permitted (i.e., schools). Due to the timing of our data collection, we were not able to collect pre-post data; however, we anticipate that we would have observed similar findings to previous studies that observed positive changes in physical activity following participation in BOKS programming [35]. 

In addition to protecting physical activity levels during the pandemic, survey results suggested that the BOKS programming may have also had a protective effect on children’s physical activity enjoyment and physical literacy. Children in our study reported moderate physical activity enjoyment, and their scores were similar to those reported by girls following participation in a different physically active after-school program [36]. On the PLAYself questionnaire, children’s reported self-assessments of their physical literacy were only slightly lower than those reported by Canadian children and youth pre-pandemic [25,37]. In addition, all reported well-being measures in the NIH-PROMIS were either within or slightly below normal limits, similar to those reported by parents in this study [14]. These quantitative findings are supported by our qualitative findings that BOKS supported children’s emotional and psychological well-being and boosted children’s confidence, particularly during the pandemic. It is well documented that physical activity participation supports the mental health and well-being of children and youth [38]. During the pandemic, children and youth in Nova Scotia reported that they missed extra-curricular activities, sport, and physical activity while schools were closed and were looking forward to returning to these activities once schools opened again [5]. Physical activity promotion may be particularly important during stressful events, such as the pandemic, to promote positive mental, emotional, and social well-being for children and youths. 

The benefits of the BOKS programming extended beyond child and youth physical activity, physical literacy, and well-being. The analysis of our qualitative data revealed that the BOKS programming was well-received by children, parents, and program leaders. Program leaders reported that it integrated well with existing school-based activities and that the activities were easy to follow and implement, particularly the movement bursts. Based on our results and the evaluation of the BOKS programming, we have several recommendations for future implementation of BOKS programming. First, there is an opportunity to build on the inclusivity of the programming to ensure activities are simpler for younger children, more challenging for older children, and more inclusive to children of all abilities by having fewer structured exercises and more games. In addition, future updates to the programming should place an increased emphasis on fun and enjoyment to ensure children are more engaged when participating. Program leaders could also further benefit from training to increase their knowledge, confidence, and skills to deliver engaging BOKS programming. 

This study reports on the program evaluation of the BOKS programming in an existing after-school program and provides important findings for future implementation of BOKS in other after-school programs. Strengths of our study include the inclusion of both child-reported, valid measures of physical activity, physical literacy, and physical activity enjoyment and qualitative interviews with parents, children, and program leaders. We were limited by some pandemic-related restrictions, and our assessors were not able to conduct in-person assessments of children and youth and were limited to surveys and interviews at only one timepoint after the completion of BOKS programming. The lack of in-person testing limited our ability to recruit and resulted in a smaller sample size than desired. We were also limited by the timeframe of the study, as the semester ended and multiple requests had already been sent to parents for their participation. It was also not possible to determine if data saturation was reached, as interviews were analyzed after data collection was complete. In the future, program evaluation of BOKS programming would benefit from a larger sample size, objective assessments of physical literacy (i.e., PLAYfun), and device-measured physical activity (i.e., accelerometers or pedometers) before and after implementation. 

## 5. Conclusions

Public health restrictions during the COVID-19 pandemic limited opportunities for children and youths to be physically active by closing recreation facilities and cancelling sports practices and games. Interviewees shared that the BOKS after-school program was the only structured physical activity opportunity for some children and youth during the pandemic and may have had protective effects on their physical activity and well-being. Our sample reported that the BOKS programming was well-received by parents, children, and trainers and helped support children’s confidence and physical, emotional, and social well-being. These findings suggest that physical activity opportunities should be maintained during the pandemic, albeit with the appropriate public health measures to mitigate the spread of COVID-19. 

## Figures and Tables

**Table 1 ijerph-19-05640-t001:** Participants’ physical literacy, physical activity participation, physical activity enjoyment, and mental and social health results.

	Mean	Standard Deviation	Minimum	Maximum
Physical Literacy (*n* = 12)				
PLAYself	68.23	13.12	43.81	87.07
Physical Activity Participation (*n* = 14)				
PAQ-C	2.70	0.48	2.13	4.03
Physical Activity Enjoyment (*n* = 11)				
PACES	4.22	0.58	3.19	4.94
NIH-PROMIS Raw Scores (*n* = 11)				
Life Satisfaction	32.10	5.04	26	40
Family Relationships	18.00	2.10	13	20
Cognitive Function	25.36	5.68	14	31
Positive Affect	16.36	3.70	11	20
Peer Relationships	32.10	5.04	26	40
NIH-PROMIS Adjusted T-Scores				
Life Satisfaction	47.73	7.39	37.4	59.5
Family Relationships	41.21	3.58	32.4	44.7
Cognitive Function	45.60	5.48	35.11	51.54
Positive Affect	51.48	10.64	37.40	63.00
Peer Relationships	48.28	8.31	39.82	64.44

Note: PLAY: Physical Literacy Assessment for Youth; PAQ-C: Physical Activity Questionnaire for Children; PACES: Physical Activity Enjoyment Scale; NIH-PROMIS: National Institute of Health Patient-Reported Outcomes Measures Information System.

## Data Availability

The dataset supporting the conclusion of this article is available from the authors upon reasonable request and the completion of a data transfer agreement.

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
