# Peer review of "The Impact of an After-School Physical Activity Program on Children’s Physical Activity and Well-Being during the COVID-19 Pandemic: A Mixed-Methods Evaluation Study"

_ijerph, 2022, doi:10.3390/ijerph19095640_

Round 1

Reviewer 1 Report

Thank you for the opportunity to review this manuscript. Please see attached comments/feedback.

Author Response

Thank you for the opportunity to review this manuscript examining the impact of an afterschool physical activity program on children’s physical activity and well-being during the pandemic. This was an interesting paper and evaluated what appears to be a great and much-needed program for increasing children’s physical activity! (Especially at a time when children’s physical activity levels have declined drastically). The main points I want to highlight are in relation to the qualitative results and the ability to relate these positive changes in children’s physical activity and well-being to BOKS. The qualitative results are quite clear and well laid out. However, it would be very helpful if more quotes could be incorporated in order to bring that information to life and to emphasize the points being made. The authors highlight the need for different perspectives early on in the manuscript, but within the qualitative results, this doesn't really come through. It would strengthen the manuscript if quotes from children, parents, program leaders could be incorporated into the different themes to help support the conclusions being made.

Regarding the positive impact on children’s physical activity and well-being, how were the authors able to differentiate between BOKS and the regular after school programming in order to determine that it was the BOKS program that had a positive impact? It was stated that 70 minutes per week was BOKS programming, but out of what total time? Can the authors please clarify how long the after-school program was (e.g., 1 hour? 2 hours?)? Apologies if I misunderstood, but if you could clarify how you were able to determine the positive impact on PA and well-being was from the BOKS programming, that would really help. Thank you.

Thank you for taking the time to review our manuscript and for providing your suggestions and comments. We have edited the qualitative results section to include a few more quotes and perspectives. We initially did not include too many quotes to keep the results section more concise. This study was designed as a program evaluation and we did not have a control group of the regular after-school programming for comparison to the BOKS programming. Additional details about the program have been added to the Study Design section. We believe the qualitative results allowed us to determine to impact of BOKS as participants, parents, and program leaders specifically commented on the impact of the BOKS activities and games.

Line 17: Could you clarify the children’s age range or mean age here?

The children were in Grades 4-6 and this is now added to the abstract on Line 17.

Line 53: Is there a citation for this? Perhaps de Lannoy et al. 2020

Thank you, this citation has now been added to this section of the introduction.

Line 59: Can you clarify if the physical activity program was implemented to address the trend mentioned in the previous sentence?
We have now clarified that out study occurred as a natural experiment and allowed us to see how a physical activity program implemented during a time of public health restrictions impacted children’s physical activity and well-being: “The timing of our study provided a natural experiment to determine the impact of an after-school physical activity program delivered during a period of strong public health restrictions” (lines 56-58).

Line 80: Perhaps clarify why ‘particularly’ in after school programs (e.g., do more students participate in after school programs? Not enough evaluation done on after school programs)

We have now clarified the importance of studying after-school programs: “as many Canadian elementary school attend after-school programs as a form of childcare and after-school programs are a promising setting for physical activity promotion (lines 73-75).

Study Design

Line 92: It would be helpful if the sample size is mentioned somewhere in the Materials and Methods section, perhaps before the measures.

Thank you for this feedback. We have now included the sample sizes for interviews and surveys in the ‘Study Design’ paragraph of the Material and Methods section.

Lines 99-101: Is there information on response rate or how many Excel leaders were involved? Could perhaps clarify, “All Excel leaders (n=#)...”

Thank you for this comment. The invitation to participate was distribute to leaders at 31 sites; however the exact number of potential leaders is unknown. This information about the number of sites has been added to line 307.

It would be helpful to have a bit more information about the demographics of the sample. For example, Line 180: Please include the ages/mean age of the children. Or what is the breakdown of gender amongst participants?

The students who were recruited were in Grades 4-6 (average age: 9.25 years; 55% female) and this has been added to the results section (line 197).

Lines 166-177: Who/how many individuals analyzed the qualitative data? Did the authors do anything to ensure rigour in their analysis? If authors could expand a bit on these points it would add strength to this section.

Thank you for this feedback. We have now added additional information about the analyses: “Analyses was completed by two of the study’s co-authors (J.B.L.Z and E.C.). To ensure rigour in their analyses, , the co-authors independently reviewed transcripts and met regularly to review content and determine themes” (lines 193-195).

Table 1: Formatting – is it possible to move ‘Deviation’ down so that the word is not hyphenated? Please include number of participants from which this data are drawn from (i.e. n= #)

Yes, ‘Deviation’ has been moved to the next line and the sample size has been added to Table 1.

Qualitative results

Great themes! It would be helpful if this section was more illustrative with quotes woven in throughout that emphasize and bring to life the themes that were developed by the researchers. As the authors noted earlier in the manuscript it is important to portray different perspectives including the child, parent, and program leader. It would be helpful if there are quotes from all of these participants in your results section.

Thank you for this feedback. We have now added additional quotes and perspectives throughout the qualitative results section to further emphasize the different perspectives.

Lines 205-211: Did children feel the same – that they were more active?

Please see lines 228-233 for students’ comments about their physical activity levels.

Lines 225-226: Can you please clarify this sentence “... one parent was pleasantly surprised that it had permitted her overtly shy child to participate in the program.” Does this mean that the child was confident enough to participate in BOKS? Or that the program helped the child feel at ease or confident enough to participate? Perhaps clarify the ‘it’ in the sentence.

This sentence has now been clarified: “One parent was pleasantly surprised to learn about the non-competitive nature of BOKS as it had supported their overtly shy child to participate in the program” (lines 271-273).

Can you please clarify why interviews were completed with the number the participants reported and how did the authors determine no further interviews were needed? (I think this is addressed in the discussion (e.g., due to pandemic) but would be helpful if reported in methods where authors discuss sample size/demographics).

We understand our sample is smaller than desired; however, we were limited by the timeframe of the study as the semester, and consequently the BOKS programming, ended and multiple requests had already been distributed to parents for their participation. Due to public health restrictions, we were not able to visit programs in-person for recruitment and depended only on email communication to parents. A comment on this has been added to the methods on lines 109-112: “The number of interview participants was limited by the timing of the study as the semester had ended and the BOKS programming was completed and by the COVID-19 pandemic as in-person recruitment and data collection were not possible.”

Discussion

Line 245: The purpose here states “to determine the impact of BOKS programming implemented in a before and after-school program...” The title of this paper only states ‘after-school program’. Please be consistent throughout the manuscript regarding whether this is both before/after or just an after-school program.

Thank for noticing this error. The text has been updated to reflect that we assessed the impact in an after-school program only.

Line 279: Perhaps change ‘that’ to ‘than’

Thank you for catching this, we have updated the text on line 290.

Author Response

Thank you for the opportunity to review this evaluation of the Build Our Kids Success (BOKS) after- school program in Nova Scotia during the COVID-19 pandemic. Children, parents, and program leaders participated in this evaluation study. The evaluation is stated to be a mixed methods design. The program is clearly important to the community.

Thank you for taking the time to review our manuscript and to provide your comments and feedback. Please find our responses to specific comments below.

Abstract
Lines 19-20: Please specify which NIH-PROMIS scales were used.
Thank you for this comment; however, we were unable to include the scales in the methods of the abstract as we were required to shorten the word count to meet the journal’s requirements. The specific scales are listed in the results section of the abstract.  

Introduction

Lines 43-45: Consider pointing out a few relevant health benefits of physical activity in children, even though they are well-established.

We have now indicated that participation I physical activity for children and youth has “physical, psychological/social, and cognitive health benefits…” (Lines 48-49).

Lines 54-56: Consider removing the comparison of PA with children in Quebec. It doesn’t add much to the argument and briefly distracts the reader.

This sentence has now been removed from the introduction.

Materials and methods:
Lines 93-94: This statement may not be needed. It is unclear at this point in the manuscript what modifications occurred and why only the measures of children’s physical literacy were included in the original protocol. Consider revising for clarity and/or moving this statement to a later part of the manuscript when discussing procedures. It seems out of place since physical literacy hasn’t been introduced yet (beyond the abstract).

Thank you for this comment. We have now moved this assessment to the ‘Quantitative Measures’ section of the Materials and Methods section.

2.1. Study design:
More details about the BOKS program are needed (e.g., How many weeks did the children participate in the after-school program? Did children sign up for a whole semester or was it week by week? How many children were in the program at a single time? What lesson topics were covered?)

This has been clarified to give further details to the quality and quantity of BOKS that children received at each site: “Although children were registered for Excel for the duration of the semester, not all children were in attendance each day. Inclusion criteria for the study included Excel program leaders, parents, and children in grades 4 to 6 who were participating in at least one session of BOKS per week. Program leaders reported that BOKS programming was delivered, on average, 70 minutes per week, with an average of 20 children in attendance per site” (lines 95-113)

Include more details regarding recruitment methods (e.g., Were the participants of this study a subset recruited from Caldwell et al. (2022)?)

Recruitment was done via email, an invitation was sent to all EXCEL families, and leaders: “Recruitment for all surveys and interviews was done via email invitation from the Excel Management to all families registered in the Excel after-school program at 31 sites” (lines 113-114).  

(3) Include a description of the inclusion/exclusion criteria and more information about the sampling frame (e.g., Elementary schools? Range of potential ages?).

Inclusion criteria has been added: “Inclusion criteria for the study included Excel program leaders, parents, and children in grades 4 to 6 who were participating in at least one session of BOKS per week” (lines 109-111).

(4) How many parents and program leaders were recruited? (This information appears in the abstract, but I am not seeing it in the body of the manuscript.)

The number of program leaders and parents who completed interviews has now been added to the Study Design paragraph.

(5) A summary of who (children, parents, program leaders) completed which measures (quantitative questionnaires versus qualitative interviews) and when may be helpful here. While this information is included under 2.2 and 2.8, an overall summary here would provide a better understanding of the overall study design before the reader begins to learn about the details of each measure.

The number of children, parents, and program leaders has now been clarified in the Study Design paragraph (lines 116-117).

2.5 Physical literacy:
Please define physical literacy here.

The definition of physical literacy has now been added.

  1. Results
    Sample characteristics (age, demographics) are missing. Please add.

Thank you for this comment. This has now been added to the results section on line 198.

3.2 Qualitative measures:
(1) The discussion of resulting themes would benefit from the integration of more supportive quotations for each theme. The results lack adequate depth.

Thank you for this comment. We initially included less quotes to keep the results section concise; however, additional quotes and perspectives have been added throughout the qualitative results section.

  1. Discussion:
    Lines 244-245 states the program was a before and after-school program. Please clarify as previous statements state the current evaluation was of after-school only.

Thank for noticing this error. The text has been updated to reflect that we assessed the impact in an after-school program only.

Lines 314-315: Please include how the smaller sample size may have affected the results (e.g., was data saturation reached?)

We understand our sample is smaller than desired; however, we were limited by the timeframe of the study as the semester, and consequently the BOKS programming, ended and multiple requests had already been distributed to parents for their participation. Due to public health restrictions, we were not able to visit programs in-person for recruitment and depended only on email communication to parents. Due to the timing of data collection, the interviews were all analyzed after data collection was complete and it was not determined if data saturation was reached (lines 388-390).

Be careful not to overstate the findings. Determining the effects of the BOKS programming on children’s physical activity and well-being appears to be an over statement given the study design. In particular, the quantitative data describe the physical activity and well-being of a small sample of children who participated in the BOKS program but do not necessarily provide evidence in support of the effects of the program. As a result, the statements in the discussion about the potentially protective effects of the program appear to be more conjecture than conclusions that follow directly from the results of the study. Consider revising these statements keeping in mind the limitations of the study.

Thank you for the comment. We had adjusted some of the language in the discussion to ensure we are not overstating our findings. For example, see lines 395 and 398.
The introduction and discussion may be improved by the inclusion of evidence from other after- school PA programs during the pandemic, if available.

We agree; however, our literature searches have not identified any articles that report on after-school PA programs implemented during the COVID-19 pandemic. We hope to see more research on this topic in the future.

Reviewer 3 Report

Review the document. 

Author Response

The research is interesting from the perspective that a non-pharmacological element is used to combat the COVID-19 virus, such as physical activity. The article provides interesting information on the impact of physical activity on children's bio-psycho-social health and on qualitative aspects on their parents and teachers in charge of the BOKS program. A limitation is the small number of the sample, but it could be avoided due to the qualitative nature of the analysis of the information based on interviews, in which the number of participants is usually small.

Thank you for taking the time to review our paper and provide your comments and feedback. We acknowledge our small sample size as a limitation of our study; however, believe our mixed-methods approach strengthens our study as we report on both the quantitative and qualitative impact of the BOKS programming.

  1. The summary exceeds 200 words, it must be reduced.

The summary has now been updated and is 200 words.

  1. Does the PLAYself questionnaire in its original version have to be administered by an evaluator? can it be self-administered? Was this bias for the interpretation of results? What can the authors answer about this (line 93 – 96).

The PLAYself questionnaire can be completed by children independently from age 7 and up. This clarification has been added to the text to supplement the information provided about PLAYself’s internal consistency, test-retest reliability and validity.

  1. Were the interviews conducted validated? were they drawn from the literature? Please clarify this (line 155).

The interview questions were not validated as it not always common practice to validate interview questions. The questions were developed by the study investigator’s based on the program evaluation’s research questions (lines 177-178).

  1. How many children? The 7 children in this study? This should be clarified (line 195).

Thank you for this feedback. We have now included the sample sizes for interviews and surveys in the ‘Study Design’ paragraph of the Material and Methods section.

  1. How many parents? This should be clarified (line 200).

Thank you for this feedback. We have now included the sample sizes for interviews and surveys in the ‘Study Design’ paragraph of the Material and Methods section.

  1. How many program leaders? This should be clarified (line 234).

Thank you for this feedback. We have now included the sample sizes for interviews and surveys in the ‘Study Design’ paragraph of the Material and Methods section.

  1. These sentences must be emphasized as study strengths (308 – 310).

Thank you, we have updated the paragraph to further clarify these sentences as study strengths.

Round 2

Reviewer 2 Report

Thank you for your revisions. In the abstract, I still recommend specifying which PROMIS scale was used.